# Cross-Modal Representational Alignment with LLM Priors for Image Generation

**Mykola Vysotskyi**
SoftServe
Sadova St, 2D, Lviv, Lviv Oblast, 79021
mvysot@softserveinc.com

**Zahar Kohut**
SoftServe
Sadova St, 2D, Lviv, Lviv Oblast, 79021
zkohu@softserveinc.com

**Anna-Alina Bondarets**
SoftServe
Sadova St, 2D, Lviv, Lviv Oblast, 79021
anbondaret@softserveinc.com

**Taras Rumezhak**
SoftServe
Sadova St, 2D, Lviv, Lviv Oblast, 79021
trume@softserveinc.com

**Volodymyr Karpiv**
SoftServe
Sadova St, 2D, Lviv, Lviv Oblast, 79021
vkarpi@softserveinc.com

## Abstract

Prior works have investigated the integration of large language models (LLMs) with rectified flow for image synthesis, but systematic studies of remain scarce. In this study, we examine how controlling the interaction between stochastic and semantic inputs during encoding, while integrating them during decoding, influences the alignment between noised latents and LLM hidden states. Our investigation shows that architectural refinements, such as dual-stream encoding and single-stream decoding, can accelerate training and improve image quality relative to LLM-adapted rectified flow baselines by enhancing representational similarity between text and visual domains. We evaluate our approach on standard image benchmarks and observe gains in both training speed and output detail preservation, indicating that structural choices in the integration of LLM features matter for cross-modal representational alignment in generative modeling. Beyond empirical improvements, our findings contribute to understanding how foundation models trained on text can develop representations that align with visual domains, revealing insights into the emergence of similar representational structures across distinct modalities. These results highlight a promising direction at the intersection of LLMs, rectified flow, and cross-modal representational analysis and motivate further explorations into unified representation learning.

## 1 Introduction

Recent advances in multimodal image generation have demonstrated the potential of Large Language Models (LLMs) to process and synthesize complex visual data. Leveraging them in text-to-image generation enhances image quality and alignment with textual descriptions. The integration of LLMs improves the representation of text, resulting in a more accurate image synthesis. This approach also allows high-quality images to be generated with fewer training data and computational resources. Liu et al. [2024]

Preprint.

The JanusFlow model Ma et al. [2024] combines understanding and generation capabilities within a single framework, using a shared Large Language Model backbone. While effective for unified multimodal tasks, its design allocates capacity to both text and image generation. In this work, we focus exclusively on image synthesis, adapting the LLMs backbone as a high-capacity semantic prior for text–image alignment. By removing the text-generation pathway, we dedicate the model's capacity entirely to improving visual quality and prompt adherence, leveraging the pretrained LLM's broad knowledge and representational strength solely for guiding the image generation process.

Specifically, we introduce a dual-stream encoder that keeps noise and text in distinct pathways but retains inter-stream communication through shared attention layers, limiting interference between stochastic variation and semantic intent.

For the decoder, we adopt a single-stream design that unifies noise and text-conditioned representations, ensuring that semantic guidance and stochastic variation act together to shape the generative process. This integration improves coherence and alignment while preserving expressive diversity.

These adjustments allow for improved alignment between textual input and generated visual output while maintaining high-quality synthesis. By updating the encoder-decoder structure, we streamline the model to better suit the specific demands of visual generation, ultimately enhancing efficiency and output quality.

In summary, our contributions are:

- We investigate how encoder–decoder design choices, specifically limiting interference between stochastic and semantic inputs at encoding while integrating them at decoding, affect the representational alignment between LLM features and visual synthesis processes.
- We demonstrate that these design adjustments yield faster convergence and higher-quality outputs compared to existing LLM-adapted rectified flow baselines
- We show that these refinements improve the alignment of noised latents with LLM hidden states, leading to faster convergence and higher-quality, semantically coherent outputs, revealing insights into how similar representational structures emerge across distinct modalities.

## 2 Related Works

### 2.1 Diffusion Models

Denoising Diffusion Probabilistic Models (DDPMs) Ho et al. [2020] and Latent Diffusion Models (LDMs) Rombach et al. [2022] have laid the groundwork for modern image synthesis. These models, including Stable Diffusion Podell et al. [2023a], leverage compressed latent spaces to achieve efficient and high-quality image generation.

### 2.2 Text-conditioned Generation

Text conditioning is pivotal in guiding generative models towards semantically coherent outputs. CLIP Radford et al. [2021] embeddings are a standard conditioning signal, with extensions like classifier-free guidance Ho and Salimans [2022] and T2I-Adapter Mou et al. [2023] enhancing integration flexibility and control.

### 2.3 Cross-Modal Alignment

Architectural innovations such as BLIP Li et al. [2022] and Pix2Seq Chen et al. [2022] have improved the alignment between vision and language. Our work builds on these by introducing a dual-stream encoder and single-stream decoder, enhancing LLM-guided image generation through improved alignment and fidelity.

### 2.4 LLM-Driven Image Generation

The integration of LLMs in image generation pipelines enriches semantic understanding and multimodal alignment Koh et al. [2023], Dong et al. [2024]. Models like GILL Koh et al. [2023] and

DreamLLM Dong et al. [2024] exemplify this trend, aligning with our approach to leverage LLMs for enhanced image synthesis.

JanusFlow Ma et al. [2024], a recent model from DeepSeek-LLM, integrates Rectified Flow Liu et al. [2022a] with a shared LLM backbone for joint understanding and generation. While effective, its general-purpose design can limit performance when compared with concurrent works in image generation task.

# 3 Background

## 3.1 Rectified Flow

Rectified Flow Liu et al. [2022a], Lipman et al. [2023] is a generative modeling approach that learns a continuous transformation from a simple prior distribution $p_0$, typically a standard Gaussian $\mathcal{N}(0, I)$, to the target data distribution using an ordinary differential equation (ODE).

Rectified Flow models the transformation of continuous $d$-dimensional data points $\mathbf{x} = (x_1, ..., x_d)$, which follow an unknown distribution $p_1$, by introducing a parameterized velocity field $v_\theta$ that dictates their evolution over time $t \in [0, 1]$:

$$\frac{dz_t}{dt} = v_\theta(z_t, t),$$

where $z_t = tx + (1 - t)z_0$ and $z_0 \sim p_0$

The velocity function $v_\theta$ is optimized to minimize the deviation between its predicted velocity and the true displacement direction between samples drawn from $p_0$ and $p_1$. The training objective is formulated as follows:

$$\min_\theta \mathbb{E}\left[||v_\theta(z_t, t) - (\mathbf{x} - z_0)||^2\right]$$

Once trained, image generation is performed by integrating the learned velocity field $v_\theta$ to transport a sample from $p_0$ to the target distribution $p_1$. Given an initial latent variable $z_0 \sim p_0$, corresponding data sample is obtained by solving the ODE:

$$z_1 = z_0 + \int_0^1 v_\theta^{opt}(z_t, t)dt$$

In practice, numerical integration is carried out in an iterative manner using solvers such as Euler's method:

$$z_{t+\Delta t} = z_t + v_\theta^{opt}(z_t, t)\Delta t,$$

where $\Delta t$ is a small step size. $z_t$ is progressively updated until $z_1$, approximating a sample from a desired distribution $p_1$.Esser et al. [2024]

# 4 Methodology

Our methodology is guided by the hypothesis that large language models, trained purely on text, implicitly learn representational structures that can be transferred to generative inference in other modalities. Prior work such as JanusFlow Ma et al. [2024] demonstrated that LLM hidden states can serve as a joint representational prior for both language and vision. We extend this line of exploration by asking whether alternative architectural designs — specifically, separating noise and text processing at the encoding stage but recombining them at decoding — can more effectively expose and exploit these hidden capacities. In this way, our work should be seen less as proposing a new end-to-end system, and more as investigating how foundational LLM representations can be adapted for cross-modal representational alignment in image generation.

Our approach builds upon advancements in large-scale generative models by integrating a dual-stream encoder and a single-stream decoder, leveraging useful insights from recent works on scaling vision and rectified flow transformers Dehghani et al. [2023], Esser et al. [2024]. Specifically, we modify traditional architectures to enhance both **Fréchet Inception Distance (FID)** Heusel et al. [2018] and **Contrastive Language-Image Pretraining (CLIP)** Radford et al. [2021] scores, ensuring superior image generation quality and semantic alignment.

Both encoder and decoder operate in a latent space of the pre-trained SDXL-VAE Podell et al. [2023b] to achieve higher computational efficiency.

## 4.1 Architectural Transition

In contrast to reliance on ConvNeXt-based encoders and decoders Liu et al. [2022b], our revised architecture (see Fig. 3, Appx. A) employs a more scalable design for high-fidelity generation. Dehghani et al. [2023], Esser et al. [2024]. The key changes include:

- Adopting a dual-stream encoder to improve feature separation and representation quality.Esser et al. [2024]
- Implementing a single-stream decoder that enhances semantic alignment through text-conditioned decoding.Dehghani et al. [2023]
- Transitioning from convolutional architectures to transformer-based layers, ensuring improved scalability and expressive power.

## 4.2 Dual-Stream Encoding

Dual-stream encoder separates the processing of textual and noise information, improving representation learning and generation fidelity. This consists of two distinct streams:

- **Noise Stream**: Encodes the random latent variables, capturing stochastic variations crucial for image synthesis.
- **Text Stream**: Encodes input textual prompts separately, preserving their structural and semantic information.

The two streams are later merged via a learned cross-attention mechanism and passed into the Large Language Model (LLM), ensuring a controlled and context-aware generative process.

This design follows prior evidence that text and image (or noise) embeddings differ substantially in their representational structure and are better handled by distinct parameterizations. In practice, this corresponds to processing each modality with separate streams, while still allowing them to interact through shared attention layers. Such separation enables each stream to maintain its own representational integrity, reducing interference and yielding more stable mappings in the rectified flow setting. Esser et al. [2024]

The encoder architecture, presented on (see Fig. 1, Appx. A), realizes the dual-stream design introduced above.

## 4.3 LLM

We initialize the semantic prior from an open-source 1.1B parameter LLM (TinyLlama-1.1B Zhang et al. [2024], Apache-2.0 license) as its backbone, utilizing its knowledge to enhance text-conditioned image synthesis.

## 4.4 Single-Stream Decoding

In the original setup, the output decoder only processed noise tokens to generate images. However, we introduce a single-stream decoder that also integrates text-informed outputs, reinforcing semantic consistency. The single-stream decoder:

1. Processes a unified representation where noise and text-conditioned outputs are fused.
2. Learns to incorporate text-driven refinements into the image generation process, improving coherence.
3. Retains stochastic expressiveness from the noise stream while ensuring adherence to textual guidance.

This fusion step operationalizes the idea that semantic and stochastic factors should ultimately converge on a single generative path. Whereas dual-stream encoders emphasize disentanglement,

a single decoding stream ensures that final outputs reflect both variability and semantic precision without misalignment. In this sense, our methodology investigates how LLM-driven representations can balance randomness and structure in cross-modal representational alignment for image synthesis.

A detailed implementation of the single-stream decoder is provided in (see Fig. 2, Appx. A).

The motivation for this transition is theoretical as well as empirical. A dual-stream encoder enables probabilistic and representational disentanglement: noise tokens model stochastic variability, while text tokens preserve structured semantic intent. Esser et al. [2024] Combination in a single-stream decoder then sustains coherence by enforcing a shared generative trajectory.

Overall, it is a structural intervention that probes whether LLM latent representations can better guide visual inference when the stochastic and semantic signals are first treated independently.

# 5 Training

The model learns to transform random noise into images, conditioned on text descriptions, by optimizing both encoder-decoder and LLM components.

## 5.1 Training objective

The training objective focuses on predicting the velocity of the latent variable transformation during the generation process. At each training step, we sample a timestep $t$ from a logit-normal distribution, and the model attempts to predict the velocity of the transformation, which guides the latent variable's movement toward the target image. The latent variable $z_t$ is computed as a linear interpolation between the initial latent variable $z_0$ and the target image $\mathbf{x}$, as follows:

$$z_t = tx + (1-t)z_0$$

where $t$ is drawn at each step and controls the interpolation between the two extremes. This means that as $t$ progresses from 0 to 1, the model gradually refines the latent variable toward a more accurate representation of the target image.

The objective function of the model can be expressed as:

$$\min_\theta \mathbb{E}\left[||v_\theta(z_t, t) - (\mathbf{x} - z_0)||^2\right]$$

where $v_\theta(z_t, t)$ represents the model's prediction of the velocity at timestep $t$ and the term $(\mathbf{x} - z_0)$ is the direction from the initial latent image to the target image.

By sampling different values of $t$ throughout training, the model learns to predict the correct velocity at each timestep, gradually transforming the latent variable $z_t$ to closely match the target image. This training process allows the model to learn how to efficiently navigate the latent space, ensuring accurate image generation aligned with the input text descriptions.

## 5.2 Training stages

We employ a two-stage training process (see Fig. 4, Appx. B) for our model.

**Stage 1.** We focus on aligning the encoder-decoder architecture with the latent space of the pretrained LLM. During this stage, we do not train the LLM itself; instead, we optimize the encoder-decoder components to match the representation space of the LLM. This helps in establishing a strong foundation for the subsequent image generation process.

**Stage 2.** Then we train the entire model, including the backbone. The LLM is fine-tuned specifically for image generation, allowing the encoder-decoder components to leverage the enhanced representation capabilities of the LLM. This stage enables the full potential of the LLM to improve image quality and alignment with textual descriptions, resulting in a fully integrated model optimized for high-quality image synthesis.

# 6 Experiment

In this section, we present the experimental setup and results for evaluating the performance of our image generation model. We assess its capabilities in generating high-quality, diverse images across various domains, evaluated using standard metrics.

## 6.1 Experiment setup and training data

We train on **3M** image–text pairs drawn from two sources: *DALL-E 3 High-Quality Captions* Egan et al. [2024] (MIT) and a curated high-aesthetic *LAION–COCO* subset (Apache-2.0), mixed at a **1:2** ratio. Images are center-cropped and resized to **384×384**; captions are used as provided. Dataset provenance, benchmark prompt isolation, and leakage prevention are detailed in Appendix F.

Training uses **2×8 H100** GPUs with PyTorch DDP, following a two-stage schedule; the full configuration (optimizer, learning rates, batch sizes, schedules) (see Table 1, Appx. B) and stage overview (see Fig. 4, Appx. B) are in the Appendix B. The end-to-end run time is ∼**8.5 days**.

## 6.2 Evaluation

For evaluation, we assess the model's performance using two key metrics: **CLIP Similarity** Radford et al. [2021], **Fréchet Inception Distance (FID)** Heusel et al. [2018]. These metrics are computed every 12,000 training iterations on a **MJHQ FID-30k** Li et al. [2024a] that is common for text-to-image models benchmarking. CLIP-ViT-Large-Patch/14 version of CLIP was used. We compare:

- the original JanusFlow 1.3B model Ma et al. [2024]. We compare performance with this model, showing that our training is fair and correct.
- baseline that leverages pretrained LLM and freshly initialized encoder-decoder proposed in Ma et al. [2024].
- a modified version with a proposed single-stream decoder,
- a modified version with a proposed dual-stream encoder,
- a fully refined model incorporating both encoder and decoder modifications.
- as an additional baseline, we trained a text-conditioned U-ViT Bao et al. [2023] scaled to approximately 1.3B parameters, following the scaling recipe suggested in *Efficient Scaling of Diffusion Transformers for Text-to-Image Generation* Li et al. [2024b].

This benchmarking provides insight into the effects of architectural modifications on image quality and text alignment. All models are compute-matched (∼1.3B parameters, same training data), except JanusFlow, used only for calibration.

See Appendices F and G for more details on the datasets and key evaluation metrics accordingly.

In addition to quantitative evaluation, we present qualitative results, showcasing generated images for a set of complex prompts. These examples highlight the model's ability to capture fine-grained details, maintain spatial awareness, and accurately represent multiobject compositions. By comparing outputs from different models, we visually assess improvements in coherence, object interaction, and adherence to textual descriptions.

## 6.3 Quantitative results

Our quantitative results indicate that the proposed modifications significantly accelerate convergence across all evaluation metrics. As shown in (Figure 1b), our model achieves higher CLIP similarity with fewer training steps compared to the baseline, demonstrating improved text-image alignment. In terms of image quality, (Figure 1a) shows a faster reduction in FID and confirming improvements in realism and diversity.
Upon completion of training, our model achieves CLIP similarity of 24.58, and 10.04 FID, as detailed in (Table 1), further validating the effectiveness of refining both the encoder and decoder.

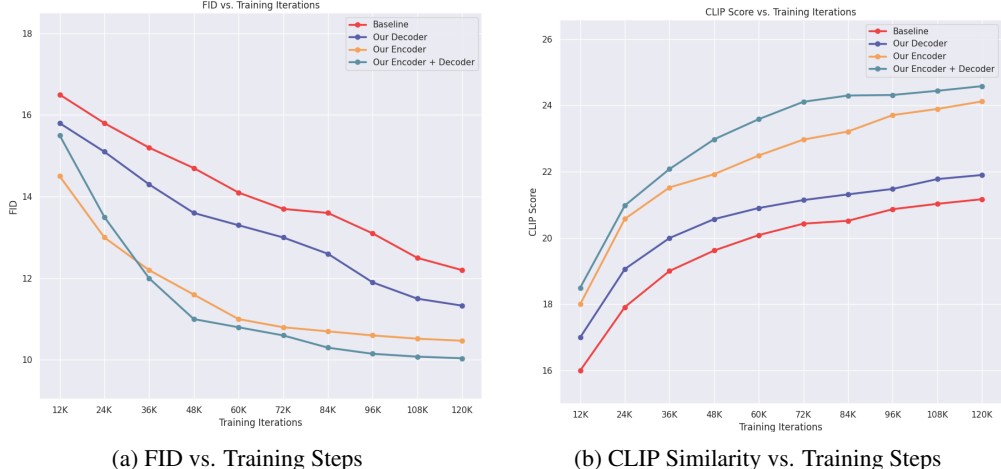

| (a) FID vs. Training Steps | (b) CLIP Similarity vs. Training Steps |

Figure 1: Quantitative evaluations across different model configurations. (a) FID vs. Training Steps. (b) CLIP Similarity vs. Training Steps.

| Model | FID $\downarrow$ | CLIP $\uparrow$ |
|---|---|---|
| JanusFlow | $9.51 \pm 0.12$ | $26.02 \pm 0.08$ |
| Baseline | $11.99 \pm 0.21$ | $21.17 \pm 0.15$ |
| U-ViT-1.3B (ours) | $10.92 \pm 0.20$ | $23.27 \pm 0.13$ |
| Changed Decoder | $11.33 \pm 0.18$ | $21.90 \pm 0.10$ |
| Changed Encoder | $10.47 \pm 0.14$ | $24.12 \pm 0.12$ |
| Changed both Encoder and Decoder | $10.04 \pm 0.11$ | $24.58 \pm 0.09$ |

Table 1: Quantitative comparison of different model configurations (after all 120k training steps). Higher CLIP Similarity indicates better text-image alignment, while lower FID represents improved image quality. Arrows indicate whether lower ($\downarrow$) or higher ($\uparrow$) values are better. Values are reported as mean $\pm$ standard deviation across 5 random seeds. **Note:** CLIP score reported at x100 scale.

## 6.4 Qualitative results

The qualitative results, presented in Figure 6 (Appx. D), illustrate the clear improvements in the performance of our model over the baseline. Our model demonstrates enhanced spatial awareness, ensuring that objects are positioned more accurately and consistently within the scene. In addition, it is demonstrating a refined understanding of object shapes. There is also a notable improvement in prompt adherence, with the generated images aligning more closely with the provided textual descriptions. These advancements lead to higher-quality images that are more realistic and true to the input prompts, showcasing the effectiveness of our approach.

We also conduct a user study in (Figure 4) to compare the results generated from the modified model (enhanced both encoder and decoder) with the baseline. We produced a dataset of 40 images, and 52 users were asked to select the best model based on two criteria: visual aesthetics and prompt following. To evaluate human preference, we asked the reviewers to answer questions about each pair of generated images with two models. More details on this user study are given in the Appendix E.

## 7 Representational Analysis

We use linear CKA to quantify alignment across two complementary experiments and report full details in Appx. H.

**Common setup.** Unless noted otherwise: MJHQ-30k, random 2k-prompt subset; Euler ODE solver with 30 steps; classifier-free guidance (CFG) = 2.0; $S$=5 seeds; linear CKA with global average

pooling across tokens before similarity; mean±1 SE across seeds. See Appx. H for definitions, aggregation and controls.

## 7.1 Experiment 1: Encoder–LLM alignment via CKA

We test whether the *dual-stream encoder* improves semantic alignment to the LLM prior by computing linear CKA between (i) the **text stream** and LLM hidden states and (ii) the **noise/semantic stream** and the same LLM states. We evaluate at multiple LLM layers $\ell \in \{2, 12, 22\}$ and rectified-flow timestamps $t \in \{0.3, 0.6, 0.9\}$.

We assume that disentangling stochastic and semantic signals at encoding reduces interference and encourages *structured alignment* to the LLM prior, especially at mid-depth layers where global and local semantics cohere.

Results show (i) consistently stronger alignment for the text stream, maximal at $\ell{=}12$, $t{=}0.6$, with positive deltas across all evaluated depths and steps, and (ii) gains for noise tokens that grow toward late time, largest at $t{=}0.9$ and most pronounced at $\ell{=}22$. Error bars across five seeds are narrow, indicating a stable effect. Table 2 reports $\Delta$CKA (Ours−Baseline) by layer and timestamp. Taken together, the layer–time trends point to text features consolidating at mid depth/step and noise features becoming increasingly structured toward late steps.

| **Text stream** vs LLM layer | $t{=}0.3$ | $t{=}0.6$ | $t{=}0.9$ |
|---|---|---|---|
| $\ell{=}2$ | $0.032 \pm 0.008$ | $0.051 \pm 0.010$ | $0.043 \pm 0.009$ |
| $\ell{=}12$ | $0.061 \pm 0.011$ | $\mathbf{0.081 \pm 0.012}$ | $0.067 \pm 0.010$ |
| $\ell{=}22$ | $0.049 \pm 0.009$ | $0.072 \pm 0.011$ | $0.058 \pm 0.010$ |
| **Noise/semantic stream** vs LLM layer | $t{=}0.3$ | $t{=}0.6$ | $t{=}0.9$ |
| $\ell{=}2$ | $0.012 \pm 0.006$ | $0.021 \pm 0.007$ | $0.038 \pm 0.008$ |
| $\ell{=}12$ | $0.016 \pm 0.006$ | $0.028 \pm 0.007$ | $0.049 \pm 0.009$ |
| $\ell{=}22$ | $0.014 \pm 0.006$ | $0.033 \pm 0.008$ | $\mathbf{0.060 \pm 0.009}$ |

Table 2: $\Delta$ **CKA (Ours−Baseline) mean±1 std over 5 seeds on 2k MJHQ-30k.** Positive values indicate better alignment with the LLM.

## 7.2 Experiment 2: Decoder-induced alignment dynamics

Using the same trained dual-stream encoder for both systems, we compare sampling with the original ConvNeXt decoder vs. our single-stream decoder. At each solver timestep $t_k$, we log encoder outputs $E^{\text{text}}(t_k)$ and $E^{\text{noise}}(t_k)$ (layer-normalized, pooled features), compute per-prompt CKA (text vs. noise), and average across prompts and seeds. We summarize the CKA–time curves by (i) AUC (higher is better) and (ii) time-to-CKA$\geq$0.5 (lower is better).

Our hypothesis is that under rectified-flow inference, the decoder's learned velocity field shapes $x_t$; a decoder that better fuses semantic and stochastic information should induce earlier and stronger text–noise coordination upstream in the encoder features.

As shown in Figure 2, our decoder produces a faster rise in text–noise alignment across solver steps and a higher terminal CKA value. Figure 3 summarizes the effect via (i) AUC of the CKA–time curve (higher is better) and (ii) time-to-CKA$\geq$0.5 (lower is better). Because the encoder is identical in both conditions, differences arise from decoder-induced velocity fields that shape the trajectory $x_t$ during rectified-flow integration. These dynamics are consistent with our quantitative gains over the re-implemented baseline (Table 1) and the faster convergence trends (Figure 1).

# 8 Discussion

Across analyses, our results indicate that structured pathways encourage convergent representational structure between LLM features and visual latents, with decoder-side integration serving as a practical unification point that translates alignment into measurable gains.

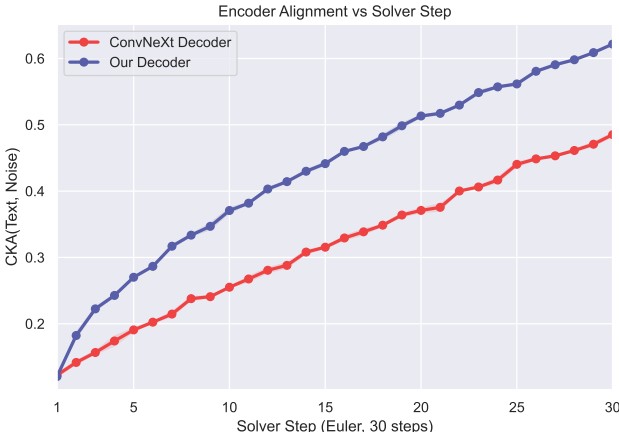

Figure 2: Encoder text–noise alignment (CKA) vs. solver step for two decoders (same dual-stream encoder). Settings follow the common protocol: MJHQ 2,000-prompt subset, Euler 30 steps, CFG= 2.0, identical prompts/seeds per seed.

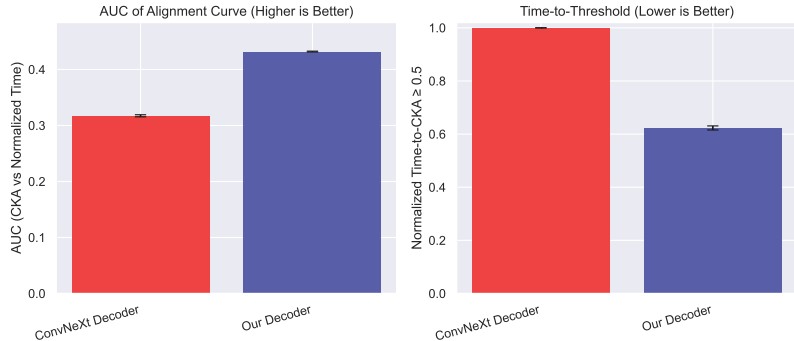

Figure 3: Summary of decoder effects. Left: AUC of CKA over solver steps (higher is better). Right: time-to-CKA$\geq$0.5 in normalized time (lower is better). Error bars denote $\pm$1 SE across five seeds. Same setup as Fig. 2.

**Reference model and scope.** We include *JanusFlow* only to calibrate our implementation and to show that our reproduction falls within the expected performance range. As already noted in the paper, JanusFlow was trained substantially longer on a much larger dataset (approximately 70M pairs) compared to our 3M-pair setup. It is therefore not a compute- or data-matched baseline.

**Ablation results Table 1.** Across metrics (FID $\downarrow$ and CLIP $\uparrow$), the structured variant with *dual-stream encoder* and *single-stream decoder* ("Both") consistently outperforms (i) our re-implemented baseline and (ii) single-change ablations ("Encoder-only" and "Decoder-only"). The encoder-side modification contributes more than the decoder-only change, indicating that separating modalities during encoding strengthens cross-modal alignment and signal disentanglement; combining both yields the strongest gains, suggesting complementary benefits that carry through to end quality.

**Learning dynamics Figure 1.** The training curves show faster convergence for the structured model: it reaches the baseline's final quality in notably fewer steps and exhibits more stable trajectories. Early-phase CLIP improvements correlate with later FID reductions, consistent with the view that isolating stochastic (noise) and semantic (text) pathways during encoding produces cleaner gradients and a more sample-efficient learning signal.

**Representational alignment (Sec.7.** CKA trends support this picture: text-stream features exhibit stronger alignment to mid-depth LLM layers, while noise-stream features become increasingly organized toward later solver steps. With an identical encoder, the single-stream decoder induces earlier

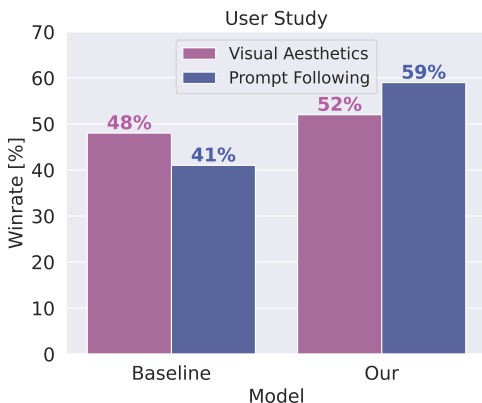

Figure 4: Human evaluation against the baseline model. Our modified architecture significantly increases prompt-following while only marginally affecting visual quality.

and stronger text–noise coordination, and a higher terminal CKA, reflecting a unified representational trajectory that aligns with the observed quantitative and convergence gains.

# 9    Limitations & Societal Impact

**Limitations** Our approach shows notable improvements in image quality and training efficiency, yet it relies heavily on high-quality pre-trained LLMs. This reliance means the model's performance is contingent on the backbone's ability to accurately represent textual data, potentially limiting effectiveness across various domains. The need for extensive training data poses challenges for generalization, as the model's capabilities are constrained by dataset diversity and coverage. Additionally, the computational cost of training remains significant, especially when fine-tuning large-scale LLMs or managing larger datasets.

**Societal Impact** While our focus is on technical advancements, it is crucial to consider the broader societal implications. Enhanced prompt adherence could inadvertently increase the risks of misuse, such as the creation of deepfakes and misinformation, which threaten privacy and public trust. The model's efficiency and portability democratize access to advanced generative technologies, but they also heighten the potential for abuse. Furthermore, reliance on large datasets and pre-trained models risks perpetuating existing biases (e.g., stereotypes in gender/occupation), as these models may reinforce stereotypes present in the training data. To mitigate potential misuse, we have decided not to release trained checkpoints, thereby controlling distribution and reducing the likelihood of bias propagation. We encourage the community to continue evaluating and addressing these ethical concerns to foster the responsible use of AI technologies.

# 10    Conclusion

In this study, we have presented a novel approach to cross-modal representational alignment by leveraging Large Language Models (LLMs) as structured semantic guides within a refined encoder-decoder architecture. By introducing a dual-stream encoder and a single-stream decoder, we effectively separate and integrate stochastic and semantic inputs, leading to significant improvements in image quality and training efficiency through enhanced representational similarity between text and visual domains. Our empirical results demonstrate enhanced alignment with textual prompts and faster convergence compared to existing LLM-adapted rectified flow baselines. These findings underscore the importance of architectural choices in achieving cross-modal representational alignment and open new avenues for further exploration at the intersection of language and vision. Our work highlights the potential of LLMs to develop representations that align with visual domains, revealing insights into how similar representational structures emerge across distinct modalities and paving the way for future advancements in unified representation learning.

*Appendix is available in Supplementary Materials.*

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
