# OpenReview forum: "Cross-Modal Representational Alignment with LLM Priors for Image Generation"
_NeurIPS.cc/2025/Workshop/UniReps — UniReps2025_

### Official Review · Reviewer_oMja · 2025-09-15
**Engineering improvements in image synthesis quality and training efficiency but lacks the theoretical depth and neuroscientific grounding.**

**Confidence:** 4

**Review:**

This paper presents an architectural refinement to LLM-guided image generation using rectified flow, introducing a dual-stream encoder and single-stream decoder to improve cross-modal representational alignment. While the work demonstrates solid engineering improvements in image synthesis quality and training efficiency, it lacks the theoretical depth and neuroscientific grounding that would make it compelling for the UniReps workshop.

The core contribution centers on separating stochastic (noise) and semantic (text) processing streams during encoding while reunifying them during decoding. The authors argue this architectural choice improves representational alignment between LLM features and visual synthesis processes, supported by experiments showing faster convergence and higher CLIP/FID scores compared to baselines. The representational analysis using Centered Kernel Alignment (CKA) provides some insight into how text and noise streams align with different LLM layers at various timesteps, with text features showing the strongest alignment at intermediate depths and mid-flow timesteps, while noise features become increasingly structured toward later timesteps.

However, the work suffers from several significant limitations. The theoretical motivation is weak - the claim that LLMs "implicitly learn representational structures that can be transferred to generative inference in other modalities" is presented without substantial evidence. The paper lacks any connection to neuroscientific principles of cross-modal processing or theories of unified representation learning that could ground these architectural choices in biological plausibility! The CKA analysis, while methodologically sound, provides only surface-level insights into representational dynamics without connecting to deeper questions about how unified representations emerge or what computational principles govern cross-modal alignment?

The comparison against JanusFlow is unfair due to massive differences in training data (70M vs 3M pairs), making the baseline comparisons less meaningful. The representational analysis focuses on correlation-based metrics without investigating causal relationships or a mechanistic understanding of how architectural changes drive the observed improvements.

From a computational neuroscience perspective, the paper misses opportunities to connect its findings to established principles of multimodal processing in the brain. There is no discussion of how the proposed dual-stream architecture might relate to known neural pathways for integrating sensory modalities, nor any consideration of whether the observed representational dynamics mirror biological mechanisms of cross-modal binding. Would authors care to describe this in detail?

The work treats LLMs as black boxes without investigating what specific computational operations enable cross-modal transfer or how these might relate to neural computation!

**Score:**

2

**Topic Fit:**

1

---

### Official Review · Reviewer_GvrU · 2025-09-16
**The paper addresses an interesting and relevant research topic, but the methodology is poorly described, and the experimental section requires significant revision to clearly demonstrate the benefits of the proposed approach.**

**Confidence:** 3

**Review:**

This paper presents an approach to text-to-image generation using LLM priors and a rectified flow-based model. The work builds on the recently proposed JanusFlow model, introducing architectural and training modifications that yield promising performance improvements.

Strengths:
- The research topic - improving generative models using LLM priors- is timely and relevant. The authors propose a potentially promising direction by employing a dual-encoder/single-decoder architecture and leveraging representational similarity.

Weaknesses:
- The paper appears rushed in its writing, which makes the ideas, though relatively simple, difficult to follow. The overall structure could be substantially improved.
- The work is not fully self-contained: the authors assume the reader is already familiar with the details of JanusFlow, which they extend.
- Figure 13 should have been included in the main paper, and the authors could have placed stronger emphasis on JanusFlow in the related work section.
- While comparisons to a baseline are provided, the authors do not demonstrate improvements over the original JanusFlow. It is also unclear on what exactly the baseline corresponds to, since the description lacks sufficient detail. The comparison would be stronger if the authors had implemented and reported results from their own reproduction of the JanusFlow architecture/pipeline as a clear reference point.

**Score:**

2

**Topic Fit:**

3

---

### Official Review · Reviewer_SvBj · 2025-09-17

**Confidence:** 4

**Review:**

## Summary
This paper investigates how architectural choices influence the alignment between Large Language Model (LLM) priors and visual latents in text-to-image synthesis, specifically within a rectified flow framework. They use an LLM's prior knowledge to guide image synthesis, advocating that textual representations learned by LLMs should be used to guide image generation. They propose a novel architecture featuring a dual-stream encoder and a single-stream decoder. The dual-stream encoder processes stochastic (noise) and semantic (text) inputs separately to be later fused with cross-attention. The single-stream decoder then integrates these pathways to guide the generative process. Experiments show that the architecture leads to faster training convergence and improved image quality and prompt adherence compared to the baseline.

## Strengths
**Well-motivated hypothesis**: The core idea that separating noise and text processing at the encoding stage while unifying them at decoding can improve cross-modal alignment is intuitive and well-articulated.

**Thorough experimentation**: The empirical evaluation is comprehensive. The authors provide a strong ablation study that isolates the contributions of the modified encoder and decoder.

**Insightful representational analysis**: A major strength of this paper is the representational analysis in Section 7. The use of CKA to measure the alignment between encoder streams and LLM layers provides concrete evidence supporting the authors' hypothesis. The experiments showing that (i) the dual-stream encoder improves alignment with the LLM prior and (ii) the single-stream decoder induces earlier text-noise coordination are particularly compelling and elevate the paper from a purely empirical result to a more profound insight into the internals.

**Rigor and reproducibility**: The paper is well-detailed, especially in the appendices. The authors provide clear visualizations of the architecture, precise training hyperparameters, and a full description of the datasets and evaluation protocols. This transparency and attention to detail lend significant credibility to the findings.

## Weaknesses
**Novelty of architectural components**: The paper's novelty lies in the specific combination and analysis of existing architectural components, rather than the components themselves.

**Limited scale of LLM backbone**: The study uses TinyLlama-1.1B as its LLM backbone. While this is a practical choice for research, it leaves open the question of how these architectural benefits vary with smaller or larger backbones. It is possible that the need for such careful architectural guidance diminishes as the raw power of the LLM increases, or conversely, that the benefits become even more pronounced. A brief discussion on the potential scaling properties would strengthen the paper


## Questions for The Authors
**Apparent contradiction**: Line 87, you said, "Our methodology is guided by the hypothesis that large language models, trained purely on text, implicitly learn representational structures that can be transferred to generative inference in other modalities.". However, in line 123 you said "...prior evidence that text and image (or noise) embeddings differ substantially in their representational structure and are better handled by distinct parameterizations...". The two seem to contradict each other as written, although I understood the point.

**Effect of stage 1 duration**:  In stage 1 of training, you freeze the backbone and train the encoder/decoder to align with its latent space. Did you experiment with the duration of this stage? Is there a risk of the encoder/decoder "overfitting" to the frozen LLM's representations, and how did you determine the 10k step cutoff?

**Score:**

4

**Topic Fit:**

3